# Large Language Models as Recommender Systems: A Study of Popularity Bias

Jan Malte Lichtenberg*
jlichten@amazon.de
Amazon Music
Germany

Alexander Buchholz*
buchhola@amazon.de
Amazon Web Services
Germany

Pola Schwöbel*
schwobel@amazon.de
Amazon Web Services
Germany

## ABSTRACT

The issue of popularity bias—where popular items are disproportionately recommended, overshadowing less popular but potentially relevant items—remains a significant challenge in recommender systems. Recent advancements have seen the integration of general-purpose Large Language Models (LLMs) into the architecture of such systems. This integration raises concerns that it might exacerbate popularity bias, given that the LLM's training data is likely dominated by popular items. However, it simultaneously presents a novel opportunity to address the bias via prompt tuning. Our study explores this dichotomy, examining whether LLMs contribute to or can alleviate popularity bias in recommender systems. We introduce a principled way to measure popularity bias by discussing existing metrics and proposing a novel metric that fulfills a series of desiderata. Based on our new metric, we compare a simple LLM-based recommender to traditional recommender systems on a movie recommendation task. We find that the LLM recommender exhibits less popularity bias, even without any explicit mitigation.

## CCS CONCEPTS

• **Information systems → Personalization**; **Language models**; **Information retrieval diversity**; **Recommender systems**.

## KEYWORDS

Popularity Bias, Recommender Systems, Large Language Models.

## 1 INTRODUCTION

Recently, general-purpose large language models (LLMs) have achieved astonishing successes across a wide range of tasks such as summarization, information extraction and content creation. In many domains, LLMs are used as foundation models—multipurpose tools that replace task-specific machine learning models. The broad applicability of LLMs, coupled with their potential for use in intent-following or conversational recommender systems, has spurred interest in exploring their role in this domain [18, 21, 39, 54, 58]. Our work analyzes LLMs as recommender systems along a specific dimension, popularity bias.

*The authors contributed equally to this research.

Popularity bias occurs when a recommender system disproportionally surfaces popular items (see Sections 2 and 3 for details), and has been present in recommender systems for decades [8, 52]. It is still present today in applications such as modern streaming services that rely on recommender systems to guide the user's consumption and exploration behavior. The potential negative impact of this bias ranges from filter bubbles over reduced user satisfaction, unfairness towards content producers to lost economic opportunity for platform providers. Strategies to quantify, explain, and mitigate popularity bias have seen growing interest lately, see [34] for a recent survey.

Just like with standard recommender systems, popularity-biased behavior would not be unexpected for an LLM-based recommender: during training, the model has likely encountered popular content more often than lesser known content. Training data biases have been shown to propagate into model generations in other contexts such as geographical [48, 61] and gender-occupation biases [28, 37, 43]. On the other hand, LLMs also provide a new opportunity to mitigate popularity bias due to their natural language interface. We can simply *ask* the LLM to recommend more niche content. Such self-debiasing [47] has been proven effective across a range of bias-mitigation tasks [40].

With the aim of quantifying the popularity bias in LLM-based recommender systems, we face a fundamental challenge: the literature lacks a consensus on what is an appropriate metric for measuring popularity bias [34]. Therefore, our first contribution is the development of a principled framework for measuring popularity bias in recommender systems. We start by outlining a set of desiderata that a metric should satisfy, focusing on interpretability and statistical robustness. We then assess existing metrics against these desiderata and introduce a new metric that satisfies them.

To conduct our experiments, we propose a simple LLM-based recommender system, which can be built on top of any general-purpose LLM, and evaluate its popularity bias against a suite of baselines. We then mitigate the bias via prompting and study the resulting accuracy/popularity-bias trade-off.

## 2 BACKGROUND AND RELATED WORK

### 2.1 Traditional top-$k$ recommender systems

Traditional top-$k$ recommender systems (RS) serve the purpose of selecting a set of $k$ most relevant items out of a much larger content pool. Common RS such as collaborative-filtering methods [26, 42, 53] operate on *item-level*. That is, these models learn an embedding space representing each candidate item and then recommend the $k$ unseen items that are closest in embedding space to the items previously consumed by the user (item-based collaborative

filtering) or items consumed by similar users (user-based collaborative filtering). Different algorithms provide different ways of learning such embedding spaces but they are almost always trained on past user interaction data such as attributed clicks, purchases, or subscriptions.

Several research works have studied popularity bias in traditional recommender systems [1, 13, 34, 59], and have investigated possible sources. Most arguments revolve around the idea that the historical interaction data used to train the recommender system contain, almost by definition, more behavioral signals from popular items than from unpopular ones. Collaborative filtering algorithms then naturally give more weight (on average) to the "naturally more popular" items, unless explicitly avoided by the algorithm [1, 31, 46]. Once such a recommender system is deployed, the popular items receive even more exposure and thus engagement. Consequently, the behavioral data that is gathered to train future generations of the recommender system will be even more popularity biased, resulting in a vicious circle of ever-increasing popularity bias [13].

## 2.2 Large language models as top-$k$ recommender systems

LLM-based top-$k$ recommenders take a fundamentally different approach in that they are language-based and operate on a *token-level*. In their simplest form, these models generate free-form text recommendations based on a prompt that describes the recommendation task and the user. The recommendations in text form are then parsed and resolved to the respective items in a catalogue before they are shown to the user.

While such models are currently difficult to scale and use in practice due to their slow auto-regressive generation procedure and arguably have not yet reached state-of-the-art performance in traditional recommender evaluation protocols, they provide various advantages that open up entirely new possibilities for recommender systems: First, LLM-based recommenders can be built on top of (open-source) general-purpose foundational models [18, 58], reducing the need for behavioral data collection and model training. [1] Second, LLM-based recommenders can generalize across different content types of a single domain (e.g., music, video, podcast in a media streaming service) or even across domains (e.g., the *P5* model [22]). Cross-domain generalization is notoriously difficult with feature-based recommender systems because of disjoint feature sets and varying consumption across domains. Third, due to their natural-language interface, LLM-based recommenders enable novel paradigms including intent-based recommmenders [9, 41] that allow the user to formulate their complex intent in natural language (e.g., *"I want to watch a New York gangster movie featuring a female lead role"*, *"Movies that likely inspired the cinematographic style of Wes Anderson"*). In conversational recommenders [30], the initial intent-based recommendations can additionally be refined by the user through follow-up requests.

However, as for standard RS, LLMs will be tilted towards what they have learned in their training set, and are prone to predicting popular items at overly high rates, resulting in popularity bias.

---

[1]This is particularly true for the movie domain because the training data of most current LLMs is based on the internet, which includes large amounts of movie reviews, discussions, and data bases.

## 2.3 Popularity bias of recommender systems

The literature has produced an abundance of metrics formalizing different interpretations of popularity bias [1, 13, 34, 59, 60]. This diversity in metric selection, while reflective of varied study objectives and applications, poses a challenge in achieving generalizability across research findings, in particular because the normative commitments as to why a particular metric has been selected often remain unstated.

Following Klimashevskaia et al.[34], we start with the general definition of *popularity bias* as a property of a recommender system that is present "*when the recommendations provided by the system focus on popular items to the extent that they limit the value of the system or create harm for some of the involved stakeholders*" [34, p.8].

Stakeholders include the service provider (e.g., a media streaming service or a job board), the content provider (e.g., artists, labels, companies that hire), and the user of the service (e.g. music listeners, job seekers). Popularity bias can have negative effects on all three stakeholders: the user might be bored by repeatedly receiving obvious or unoriginal "mainstream" recommendations; the service provider might suffer customer churn because of their decreased engagement or recommend only items that would have been sold anyways (and thus miss out on long-tail sales [5]); and new or niche content providers might have a hard time entering or be driven out of the market due to limited exposure. Specifically, if a recommender system inordinately promotes items by a group of already popular artists, it may limit exposure of historically disadvantaged groups of content creators, hereby reinforcing this inequality [15]. This is a common concern in the fair ML literature: existing biases are picked up and exacerbated by ML-based systems [7, 51].

Note that while Klimashevskaia et al.'s definition of popularity bias does not prescribe a specific way of how the bias should be quantified, it excludes some existing definitions. For example, unlike in the popularity bias definition by Zhao et al. [60], it is acceptable for some items to be more popular than others a priori. It is equally acceptable to recommend popular items at higher rates; this would constitute popularity bias according to the definition by Abdollahpouri et al. [1] which is reminiscent of the fairness metric demographic parity [12]. Instead, we argue that recommending more popular items more often is acceptable if they indeed are more *relevant* to the user, following an interpretation similar to equality of opportunity [24]. Predicting popular items starts to "*limit the value of the system or create harm for some of the involved stakeholders*" when predicted popularities exceed the user's base popularity in expectation. We will formalize this in the following section.

## 3 QUANTIFYING POPULARITY BIAS

In working towards reducing the ambiguity of existing popularity bias measures, we propose a framework for defining a popularity bias metric for a given problem and data set. More specifically, we first define a parametrized formulation of popularity bias that generalizes several existing popularity bias metrics for specific parameter settings. We then define a set of theoretical desiderata for an interpretable and statistically robust popularity bias metric and evaluate existing metrics against these standards. Finally, we introduce a new metric that satisfies our desiderata.

As motivated in Section 2.3, we quantify popularity bias of a recommender system with respect to the user's experience. A recommender system is positively (negatively) popularity biased if it recommends popular items at higher (lower) rates than appropriate for that user.

We establish the following basic assumptions on how popularity is measured:

(1) **Data**. The data set provides raw popularity scores $\phi(a)$ for every item $a$. Raw scores are an aggregate consumption measure over all users (e.g., total number of reviews for a movie, or total number of plays for a song). Online consumption patterns tend to have heavy tails, i.e., a few items account for the major share of interactions. Empirically, item popularity follows a power law (Pareto distribution) arising from "rich get richer" dynamics, [6, 14, 44]. Power laws (and more specifically Pareto-distributions) have probability density functions of the form $p(x) \propto x^{-\alpha}$, for $x \geq x_{min} > 0$, where $\propto$ denotes proportional up to a constant. The coefficient $\alpha$ determines the tail behavior of the distribution. Power laws $\alpha \leq 2$ do not have a finite mean and for $2 < \alpha \leq 3$ the variance is infinite (and consequently all higher moments). Hence, any metric based on averages (like mean, standard deviation, variance, skew and kurtosis) are ill-defined, as empirical averages do not converge. Consequently popularity bias metrics with potentially huge fluctuations will have little scientific value. Therefore, raw popularity bias scores are often transformed implicitly by a function $g(\phi(a)) : \mathbb{R} \to \mathbb{R}$ for all items $a$. For instance, $g$ normalizes popularity scores to a certain range, or removes heavy tails. These transformations are often done without explanation, a point that we address below.

(2) **Recommender popularity**. The popularity of a top-$k$ recommendation is an aggregate popularity score of all items in the slate,

$$AggP(r) = h\left(\{g(\phi(a))\}_{a \in p_r}\right),$$

where $h : \mathbb{R}^k \mapsto \mathbb{R}$ is a function that maps scores $\{g(\phi(a))\}_{a \in p_r}$ to a single value,[2] $k = |p_r|$, and $p_r$ is the set of all items chosen by the recommender.

(3) **User popularity**. A user's popularity preference is defined by the aggregate popularity score of all (or the last $n$) items the user consumed in the past (e.g., watch history for a movie streaming service),

$$AggP(u) = h\left(\{g(\phi(a))\}_{a \in p_u}\right),$$

with $h$ as above and $p_u$ denotes the set of items a user interacted with.

To define a complete metric, one therefore has to make choices on (a) how the recommendation popularity $AggP(r)$ and user popularity preference $AggP(u)$ are aggregated on the recommender and user level and how they are combined to calculate the actual bias metric (with a slight abuse of notation, we will denote such a function by $M(u, r)$), and (b) how (if at all) the raw popularity scores are transformed by a function $g(\phi(a)) : \mathbb{R} \to \mathbb{R}$ for all items $a$. Most popularity bias studies only focus on the definition of $M$

---

[2]As an example consider the empirical average, that maps scores to one value.

and $h$, while treating $g$ independently. We argue that the properties of a popularity bias evaluation framework depend jointly on $M$, $h$ and $g$.

To make principled choices of $M$, $h$, and $g$, we define a set of desiderata for our final metric. We then discuss existing metrics and define our own metric.

## 3.1 Desiderata for a popularity bias metric

We define the following desiderata for a popularity bias metric $M(r, u)$ and raw popularity transformation function $g$.

(1) **Well behaved**. We want our metric to be *well behaved* in a statistical sense, i.e., more data leads to more stable results. Our aggregation function $h$ and transformation $g$ must be chosen such that $AggP(u)$ (and $AggP(r)$ respectively) become more precise with more data (i.e., when $|p_u|, |p_u| \to \infty$). This is crucial for reproducible and coherent statements about the popularity bias of a system. Without this requirement resulting metrics are meaningless as they measure random fluctuation in the data. This desideratum ensures *construct reliabilty* [29].

(2) **Centered around** $0$. If $AggP(r) = AggP(u)$ then $M(r, u) = 0$. If there is no difference in aggregate popularity between recommendations and baseline popularity for a user, the metric should map to $0$. This requirement influences $M$, and how we compare the sets $p_u$ and $p_r$.

(3) **Anti-symmetry**. $M(r, u) = -M(u, r)$. Popularity bias can be "positive" or "negative". While most of the literature is concerned with "positive" popularity bias, i.e., recommending too popular items, "negative" popularity bias can have detrimental effects on the relevance of recommended items. We treat "positive" and "negative" popularity bias equally. We want to discover and mitigate both directions in our experiments. This will again influence the choice of $M$.

(4) **Sensitivity to the long tail**. Our metric must be robust to large popularity values while at the same time being sensitive. The same absolute difference for large popularity values matters less than the same difference for small popularity values.
    We define two sets of recommendations $\bar{r}$ and $\underline{r}$ where the main difference is that $\bar{\cdot}$ has an overall higher popularity than then items in $\underline{\cdot}$. We keep fixed the set of user items $u$. We then want that

$$|M(\underline{r} \oplus \epsilon, u) - M(\underline{r}, u)| > |M(\bar{r} \oplus \epsilon, u) - M(\bar{r}, u)|,$$

where $\oplus$ denotes an increase of the popularity of one item by $\epsilon > 0$. A shift in popularity for high popularity items of a recommender is relatively smaller than a shift in popularity of less popular items. This will influence all choices of $M$, $h$ and $g$. This is important as a small change in the tail of the less popular items should have a relatively stronger influence than an increase in popularity of the most popular items.

(5) **Componentwise monotonicity**. We want a metric monotonic in the popularity scores. If popularity scores are increased by $\epsilon > 0$ for the recommender this should lead to the metric becoming larger (respectively smaller for an increase

on the user side). This requirements reads as

$$M(r \oplus \epsilon, u) - M(r, u) > 0,$$

using the same notation as before. This is important as the overall *level* of popularity matters, which is assured by monotonic behavior.

## 3.2 Existing metrics and our desiderata

We review a subset of existing metrics commonly used in the literature and discuss them in light of the above desiderata. We selected metrics that are consistent with our previously stated base assumptions. In particular, they can all be formalized within the same framework for specific functions $h$, $g$, and $M$, as shown in Table 1. While this list of metrics is by no means exhaustive, we think that the selected metrics are representative of the main ideas and assumptions underlying most existing metrics in the literature. Table 1 provides a condensed overview of the alignment between these metrics and our desiderata.

*Average popularity lift.* Abdollahpouri et al. [3] use the lift of group average popularity as a metric to quantify popularity bias. Their metric is based on groups of users. For the sake of simplicity, we focus on overall average popularity (i.e., assuming only one single group). Average popularity at the user level $u$ (respectively, at the recommender level $r$) is defined as

$$AP(u) = \frac{\sum_{a \in p_u} \phi(a)}{|p_u|}, \qquad AP(r) = \frac{\sum_{a \in p_r} \phi(a)}{|p_r|}.$$

Importantly, no transformation of the raw scores is performed, that is, $g$ is just the identity and the score aggregation function is given by the empirical mean. The final metric, average popularity lift, is then given by

$$\Delta AP(r, u) = \frac{AP(r) - AP(u)}{AP(u)}. \tag{1}$$

This metric, unfortunately, is at odds with several of our desiderata. Regarding Desideratum (1), note that the above metric is given as an empirical average of the samples of a random variable, that is, the raw popularity scores. Therefore, we would expect that the average converges to its expectation, i.e., $AP(u) \to \mathbf{E}\phi(a)$ as the number of samples $|p_u| = n \to \infty$. However, if raw popularity scores follow a power law, as we argue at the top of Section 3, this might not be the case; the metric might not converge to anything meaningful even with an infinite amount of data. The same reasoning is also valid for metrics that are derived from some form of average popularity, such as those proposed in [15] and [35], which use a popularity bias metric based on empirical moments of the raw scores.

Furthermore, the metric in (1) also does not satisfy the desideratum on anti-symmetry. Consider a user with $AP(u) = 0.5$ and a recommendation with $AP(r) = 1$ Then $\Delta AP(r, u) = 1$ but $\Delta AP(u, r) = -0.5$. In other words, (1) reports relative change (popularity bias in this case is 1, or 100%, because the recommendation popularity is 100% larger than the user base popularity). Although this could be fixed by omitting the numerator, this would only fix the required anti-symmetry, not the other desiderata. Finally, the metric in (1) also weighs changes of the popularity equally, regardless of the

scale, putting it at odds with our required long tail sensitivity. On the other hand the metric is monotonic, satisfying Desideratum (5).

*Gini index.* The Gini index [23] is a global measure of inequality of a distribution. For the case of popularity, we define it by considering the set of items consumed by a user and the items surfaced by a recommender as

$$Gini(r) = \sum_{i=1}^{|p_r|} \left( \frac{2i - |p_r| - 1}{|p_r|} \right) \left( \frac{\phi(a_i)}{\sum_{a_i \in p_r} \phi(a_i)} \right),$$

$$Gini(u) = \sum_{i=1}^{|p_u|} \left( \frac{2i - |p_u| - 1}{|p_u|} \right) \left( \frac{\phi(a_i)}{\sum_{a_i \in p_u} \phi(a_i)} \right).$$

For the specific purpose of this metric we assume that the items $\phi(a_i)$ are sorted in increasing order (i.e., $\phi(a_i) \le \phi(a_{i+1})$), see also [4, 11]. As these measures are defined at the level of the recommender (respectively the user) we need some way of comparing these two values. We suggest to use either some form of (relative) difference or ratio:

$$\Delta Gini(r, u) = Gini(r) - Gini(u),$$

$$\%\Delta Gini(r, u) = \frac{\Delta Gini(r, u)}{Gini(u)},$$

$$\%Gini(r, u) = \frac{Gini(r)}{Gini(u)}.$$

This comparison finally amounts to comparing the inequality of popularity in the recommender profile vs. the inequality of popularity in the user profile. Note that only the first metric $\Delta Gini(u, r)$ is anti-symmetric and zero centered, which fulfills our Desiderata (2) and (3). The metric $Gini(r)$ is well defined in case of finite variance and reasonable estimators exist in the case of infinite variance, see [20, 56], satisfying Desideratum (1). A small change in a large popularity value has smaller impact than the change of the popularity value of a small value, which fulfills our requirement of the long tail sensitivity. However, as a measure of inequality within a distribution, the Gini index is, in general, not monotonic. Imagine a case of an equal popularity of all items. If now all popularity values of the recommender are lifted by an equal increment, the Gini index remains unchanged, as the overall inequality is not varied, which is at odds with Desideratum (5).

*Popularity rank correlation.* For the rank correlation of a popularity bias metric we use the definition of [62] and adjust it to our notation. The popularity rank correlation measures how strong the popularity measured by the popularity scores $\phi(a)$ of a set of items is correlated with the rank decided by a recommender. We define PRU as

$$PRU(r, u) = SRC\left[ \{\phi(a) \in p_u \cap p_r\}, \operatorname{rank}_r(\{a \in p_u \cap p_r\}) \right],$$

where $SRC$ denotes the Spearman-rank correlation, $p_u \cap p_r$ denotes the intersection of items that are both part of the recommender profile and the user profile and $\operatorname{rank}_r(\cdot)$ is the function that assigns to every item $a$ a rank decide by the recommender $r$. Note that this definition requires a notion of ranking by the recommender which is not required by the other metrics. Additionally, this metric intersects items from the user and recommender profile, which puts it slightly at odds with our imposed framework. The metric is well-behaved as ranks can be computed without additional requirements

| Desiderata | (Group) Average popularity lift [3] | Gini index [4, 11] | Popularity rank correlation [62] | Herfindahl-Diversity [4] | Log popularity difference (ours) |
|---|---|---|---|---|---|
| (1) well behaved | ✗ | ✓ | ✓ | ✓ | ✓ |
| (2) zero centered | ✓ | ✓ | ✗ | ✓ | ✓ |
| (3) anti symmetric | ✗ | ✓ | ✗ | ✓ | ✓ |
| (4) long tail sensitivity | ✗ | ✓ | ✗ | ✗ | ✓ |
| (5) monotonicity | ✓ | ✗ | ✗ | ✗ | ✓ |
| Aggregation $h$ | mean | Gini coefficient | identity | sum of squares | mean |
| Transformation $g$ | identity | normalization by total popularity | rank | normalization by total popularity | log |
| Comparison $M$ | relative difference | difference | correlation | difference | difference |

Table 1: Overview of common popularity bias metrics. We choose the aggregation function $M$ (amongst $\Delta$, $\Delta\%$ and $\%$) such that most desiderata are satisfied, indicated by ✓.

over user and recommender profiles. However, the metric is neither zero centered, nor anti-symmetric. This metric does not satisfy the long tail sensitivity and monotonicity desiderata as changes in the value of the popularity score have no influence as long as the rank is not changed.

*Herfindhal index.* The Herfindhal-index [27] was originally developed as a measure of economic concentration and has since been applied to measure popularity dispersion, see [4]. We define it at the level of a recommender (a user respectively) as

$$H(r) = \sum_{a \in p_r} \left( \frac{\phi(a)}{\sum_{a \in p_r} \phi(a)} \right)^2, \; H(u) = \sum_{a \in p_u} \left( \frac{\phi(a)}{\sum_{a \in p_u} \phi(a)} \right)^2.$$

The metrics are again defined at the user and recommender level, therefore we need an approach for computing a measure of popularity bias based on the two. Analogous to above we identify the following form of (relative) difference or ratio:

$$\Delta H(r, u) = H(r) - H(u), \; \%\Delta H(r, u) = \frac{\Delta H(r, u)}{H(u)}, \; \%H(r, u) = \frac{H(r)}{H(u)}.$$

The metric normalizes individual popularity scores by the sum of all scores in the profile, making it well-behaved (the metric is bounded between 0 and 1). By choosing the difference of the values, we can satisfy Desiderata (2) and (3). However, large popularity values have an overall dominant effect (due to the squaring of the values), putting it at odds with Desideratum (4), Desideratum (5) (monotonicity) is also not fulfilled.

## 3.3 Properties of the log popularity difference metric

We suggest a new metric, denoted *log popularity difference*, that satisfies all our desiderata. It is conceptually similar to the average popularity lift metric but drops its normalization term (to ensure anti-symmetry) and introduces a log transformation of the raw popularity values (to ensure statistical robustness). The log popularity difference metric is defined as

$$\Delta M(r, u) = \frac{\sum_{a \in p_r} \log \phi(a)}{|p_r|} - \frac{\sum_{a \in p_u} \log \phi(a)}{|p_u|}. \quad (2)$$

For data with popularity scores $\phi(a)$ following a Pareto distribution, this metric satisfies all Desiderata: Desideratum (1): it is well-behaved because the log transformation of a Pareto-distributed random variable follows an exponential distribution with well-defined mean and variance; (2): it is centered around zero; (3): the metric is anti-symmetric; and (4): changes in the tail (for high popularity values) have a weaker impact than changes for small popularity values due to the log transformation. Finally, the metric satisfies Desideratum (5) because the log transformation is monotonic itself and thus preserves the monotonicity of the average popularity aggregation function. We will use this new metric to measure popularity bias in our experiments.

## 4 EXPERIMENTS

We aim to answer the following research questions in our experiments:

- (RQ1) How does a recommender based on off-the-shelf LLMs compare to traditional behavioral recommenders in terms of both recommendation accuracy and popularity bias ?
- (RQ2) How effective are simple prompt-based popularity bias mitigation strategies? How do these mitigation strategies trade off popularity bias against recommendation accuracy?

We conduct our experiments in the movie recommendation domain using the 'MovieLens 10M' data set [25]. The data set contains 10 million ratings on 10,000 movies from 72,000 users. We chose this dataset for the following reasons: 1) we expect general-purpose LLMs to be fairly good off-the-shelf movie recommenders, given the vast amount of movie-related content on the internet including reviews and discussions; and 2) we use the *LensKit* [17] library to evaluate standard recommendation algorithms on the task and compare popularity bias and relevance of the LLM recommender to such classical methods. We start by defining the LLM-based recommender.

## 4.1 WOrld Knowledge recommender (WOK): A simple LLM-based movie recommender

We use a pre-trained LLM with a prompt template that simply asks to recommend a list of ten movies based on the watch history of a

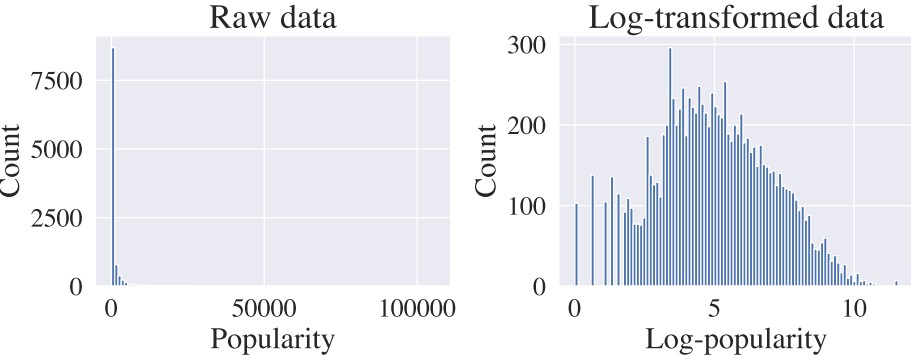

**Figure 1: Popularity scores of the MovieLens dataset. Left are the raw scores, i.e., counts of how often a movie has been rated. We run a goodness-of-fit test (using the distfit package) for several heavy tailed distributions and find that a Pareto-distribution, i.e., power law, is the best fit. The estimated coefficient is $\alpha = 0.68$, makeing both mean and variance undefined. Right are log-transformed popularity scores.**

given user. The prompt template also contains formatting instructions so that we can reliably parse the natural-language output returned by the LLM and resolve it to a list of movie identifiers present in the MovieLens 10M dataset. This is needed to calculate relevance and popularity bias of the recommendation. Furthermore, the LLMs are instructed to not recommend movies released after 2008, as these movies are not contained in the MovieLens data set, as well as movies that a user has already watched.

We call the resulting model *WOrld Knowledge recommender* (WOK) because it relies entirely on the world knowledge (also known as parametric knowledge) acquired during pre-training, i.e., it has not explicitly been trained on the MovieLens 10M dataset to recommend movies. We use various off-the-shelf LLM-APIs from Anthropic and OpenAI as back-end LLMs and call the resulting models WOK-*model-name* for different specific models during our experiments. The Appendix contains further details on API parameter choices.

### 4.2 Baselines

In order to study the relationship between predictive performance and popularity bias we chose a range of simple baselines to compare our WOK recommender with. *ItemKNN:* A traditional collaborative filtering algorithm, K-nearest neighbors based on item similarities. As recommended in the literature [16] we pick $K = 30$, i.e., we consider the 30 most similar items for each item. *UserKNN:* Traditional collaborative filtering based on K-nearest neighbors based on user similarities [50, 53]. As before we pick $K = 30$, i.e., we consider the 30 most similar users for each test user. *Top Popularity Recommender:* Always recommend the $k$ most popular items. *Random Recommender:* Recommend $k$ items uniformly at random.

### 4.3 Experimental Setup

We follow a standard evaluation protocol, where the ratings of each user are split into a training and testing set. The movies in the training set are used to train the algorithm, or in the case of the LLM recommender, as an input to the model at run-time (see Section 4.1). The recommender then generates a slate of recommendations.

To measure the recommendation accuracy we report top five hit rate (HR@5) and top ten hit rate (HR@10).

To calculate the popularity bias of different algorithms, we approximate the raw popularity score of a movie by the total number of ratings it received. The raw as well as log-normalized scores are plotted in Fig. 1. For LLM-based recommenders, we additionally report the number of invalid recommendations. Invalid recommendations occur when the LLM violates the instructions it has received via the prompt template. This could mean that the recommended item is formatted incorrectly, has already been watched by the same user in the past, is not referring to an existing movie title, or is a movie title that does indeed exist but is not part of the MovieLens data set. Note that the baseline recommenders have access to a valid candidate set and thus will always recommend valid titles.

We repeat the experiment over five folds and report the average ± the standard error of the mean in Table 2. Each fold consists of 1000 users (i.e., we subsample the original dataset).

### 4.4 Results

Figure 2 and Table 2 show the results. A perfect recommender would achieve HR@5=HR@10=1 while exhibiting zero popularity bias. Regarding predictive performance, the traditional user-based collaborative filtering algorithm (UserKNN) works best. It is also among the least popularity biased models.

The lowest popularity bias, surprisingly, is achieved by the Anthropic Claude-based WOK model (WOK-claude-v2.1). In fact, only WOK-gpt-3.5 exhibits a higher popularity bias than the least-biased baseline model. Note also that the Random recommender has a strong negative popularity bias. This indicates that the watch histories of users contained in the MovieLens data set contain movies with much higher-than-average popularity scores.

We also note the low number of invalid items for the WOK models, with all models returning valid movie recommendations in at least 8.5 out of 10 cases. This means that the LLM recommenders not only correctly format the recommended movies, but also respect the query, that is, they recommend movies from valid years and not do not recommend movies that the user has already watched.

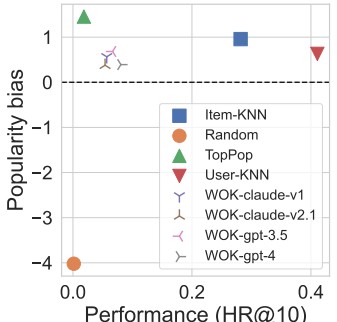

| | HR5 ↑ | HR10 ↑ | popularity bias ↓ | unmatched ↓ |
|---|---|---|---|---|
| Random | 0.001 ± 0.001 | 0.001 ± 0.001 | −4.020 ± 0.013 | 0 |
| TopPop | 0.005 ± 0.001 | 0.014 ± 0.002 | 1.455 ± 0.010 | 0 |
| ItemKNN | 0.204 ± 0.006 | 0.282 ± 0.006 | 0.957 ± 0.009 | 0 |
| UserKNN | **0.313 ± 0.007** | **0.411 ± 0.007** | 0.630 ± 0.010 | 0 |
| WOK-claude-v1 | 0.044 ± 0.003 | 0.057 ± 0.003 | 0.565 ± 0.010 | 1.360 ± 0.016 |
| WOK-claude-v2.1 | 0.042 ± 0.003 | 0.054 ± 0.003 | **0.377 ± 0.010** | 1.361 ± 0.018 |
| WOK-gpt-3.5 | 0.047 ± 0.003 | 0.067 ± 0.004 | 0.682 ± 0.010 | 0.613 ± 0.013 |
| WOK-gpt-4 | 0.050 ± 0.003 | 0.081 ± 0.004 | 0.392 ± 0.008 | 0.869 ± 0.013 |

**Figure 2 & Table 2: Results on a subsample of the MovieLens 10M dataset. We repeat the experiments** 5 **times, with** 1000 **users in each fold. Reported are the mean plus/minus one standard error of the mean.**

The gap in predictive performance of WOK models compared to UserKNN and ItemKNN can likely be explained by the additional information that those models have access to. While WOK models in this current implementation only access an individual test user's watch history, the collaborative filtering baselines have access to the full history of all user-item interactions. One could make this additional information available to the WOK models by fine-tuning them or infusing the user-item interactions via Retrieval-Augmented Generation (RAG, [36]).

## 4.5 Popularity bias mitigation and minimization via prompting

A range of mitigation strategies have recently been explored for different types of biases in LLMs. Interventions target the training data [61, 63], the learned representations [10, 45] or employ fine-tuning [48]. However, a surprisingly simple approach has been shown effective [40]. The model can be asked to 'self-debias' [47] via its prompt. In our case, we attempt to leverage the model's world knowledge regarding whether or not a movie is popular, and then ask it to focus on less popular items for debiasing. If successful, this mitigation strategy would be greatly beneficial from a usability perspective: Unlike other methods, it allows a user to configure the desired level of popularity via a natural language interface – no ML expertise required.

**Mitigation.** To evaluate this method, we add the additional instruction "*Recommend movies that match the average popularity level of the movies the user watched in the past. For instance, if the user mostly watched blockbusters, you should recommend movies that are also blockbusters. If, on the other hand, the user watched less well-known movies, you should recommend niche movies.*" to each of the WOK recommenders. Every such recommender is marked by a *-mitigate* suffix.

We use the same experimental setup as described in Section 4.3. Table 3 and Figure 3 show the results. The popularity bias of all *mitigate*-models (hollow triangles in Figure 3) is slightly decreased when compared to the respective base models ("tripods" in Figure 3). For all base-LLMs apart from claude-v1, this reduction in bias is traded off against a reduction in prediction performance.

Note with the exception of claude-v2.1, the popularity bias values remain relatively close to their base values, or in other words,

the mitigation strategy is not highly effective. We therefore also try a second, more extreme mitigation strategy to study how far we can push the WOK recommender into providing long tail recommendations.

**Minimization.** To this aim, we replace the *mitigate*-instruction by the following instruction: "*Recommend indie, niche, or less well-known movies, avoiding mainstream blockbusters.*" The resulting models are marked by a *-minimize* suffix.

The results are again shown in Table 3 and Figure 3. The *minimize* strategy does indeed reduce popularity bias to a point where the resulting models show strong negative popularity bias. However, this strategy also results in a strongly decreased recommendation accuracy across all LLMs.

Note also that the number of unmatched items increases considerably when using the mitigation prompt (except for claude-v2.1), which highlights an important limitation of this experiment. Recall that the MovieLens data sets contains "only" 10,000 movies and that WOK recommenders do not have access to this list of valid movies. As a consequence, WOK models that aim to minimize popularity bias might be punished for recommending niche movies that do not appear in the MovieLens catalogue, but could nevertheless be relevant to the user.

## 4.6 Correlation of popularity bias metrics

We compare the metrics from Table 1 with each other based on the popularity bias values that are reported in Table 2 (8 values, original model performance for each metric) and Table 3 (8 additional values, model performance under two mitigation strategies for each metric). In total we base the correlation analysis on 8+8 = 16 popularity bias values for four different metrics, i.e., Gini–diversity, Herfindahl–diversity, average popularity lift, and our suggested metric, the log popularity difference. Unsurprisingly, log popularity difference and average popularity lift show a very strong correlation, as measured by Kendal's $\tau$ [33], as shown in Figure 4. These two metrics are strongly correlated, as both metrics are very similar in nature, but average popularity lift does not satisfy the Desiderata (1), (3) and (4). For an analysis of the practical implications of this, see Appendix B.

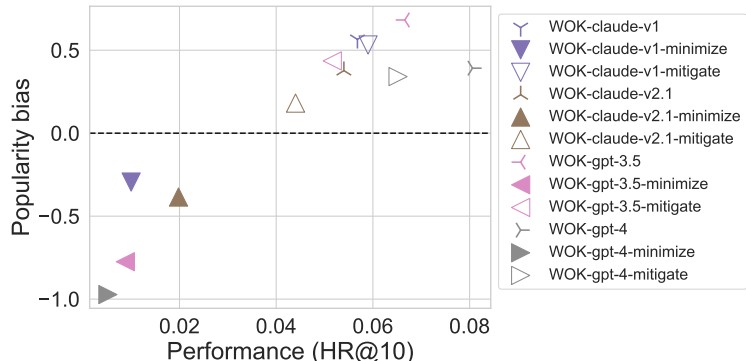

| | HR@5 ↑ | HR@10 ↑ | popularity bias ↓ | unmatched ↓ |
|---|---|---|---|---|
| WOK-claude-v1 | 0.044 ± 0.003 | 0.057 ± 0.003 | 0.565 ± 0.010 | 1.360 ± 0.016 |
| WOK-claude-v1-minimize | 0.008 ± 0.001 | 0.010 ± 0.001 | −0.295 ± 0.009 | 2.001 ± 0.031 |
| WOK-claude-v1-mitigate | 0.046 ± 0.003 | 0.059 ± 0.003 | 0.532 ± 0.010 | 1.118 ± 0.014 |
| WOK-claude-v2.1 | 0.042 ± 0.003 | 0.054 ± 0.003 | 0.377 ± 0.000 | 1.361 ± 0.018 |
| WOK-claude-v2.1-minimize | 0.015 ± 0.002 | 0.020 ± 0.002 | −0.385 ± 0.010 | 0.787 ± 0.016 |
| WOK-claude-v2.1-mitigate | 0.033 ± 0.003 | 0.044 ± 0.003 | **0.181 ±0.010** | 0.792 ± 0.012 |
| WOK-gpt-3.5 | 0.047 ± 0.003 | 0.067 ± 0.004 | 0.682 ± 0.010 | **0.613 ± 0.013** |
| WOK-gpt-3.5-minimize | 0.007 ± 0.001 | 0.009 ± 0.001 | −0.775 ± 0.013 | 3.393 ± 0.031 |
| WOK-gpt-3.5-mitigate | 0.037 ± 0.003 | 0.052 ± 0.003 | 0.438 ± 0.012 | 1.105 ± 0.022 |
| WOK-gpt-4 | **0.050 ± 0.003** | **0.081 ± 0.004** | 0.392 ± 0.008 | 0.869 ± 0.013 |
| WOK-gpt-4-minimize | 0.004 ± 0.001 | 0.005 ± 0.001 | −0.972 ± 0.009 | 1.549 ± 0.019 |
| WOK-gpt-4-mitigate | 0.045 ± 0.003 | 0.0652 ± 0.003 | 0.341 ± 0.007 | 0.914 ± 0.012 |

**Figure 3 & Table 3: Results for the mitigation experiment. The results are grouped by base LLM in the WOK model. Bold numbers indicate best performance across all models.**

Herfindahl– and Gini–diversity measure fundamentally different things than popularity lift and log popularity difference (the difference of popularity *diversity* between user and recommendation vs. the difference of *average popularity* between user and recommendation). Hence, the two families of metrics are not expected to correlate well. In practice, we observe them to be negatively correlated in Figure 4.

## 5 DISCUSSION AND FURTHER WORK

Recall that our goal was to study popularity bias in recommenders using general-purpose LLMs to evaluate their suitability as off-the-shelf recommenders, as opposed to constructing a state-of-the-art recommender in terms of recommendation accuracy. We therefore deliberately kept the LLM recommender as light-weight as possible to ensure that our findings are reflective of the intrinsic bias in LLMs, rather than being confounded by complex, model-specific factors. In doing so, we aim to provide insights that are broadly applicable across various LLM implementations, contributing to a more generalized understanding of popularity bias in these systems. Nevertheless, the recommendation accuracy of the WOK recommender could likely be improved using various techniques, including 1) fine-tuning on a training set of successful recommendations; 2) "grounding" the recommendations in a movie catalog to avoid hallucinations (via RAG, [36]); or 3) more advanced prompting techniques such as chain-of-thought prompting, or more advanced prompt optimization techniques [19, 55]. Furthermore, these or

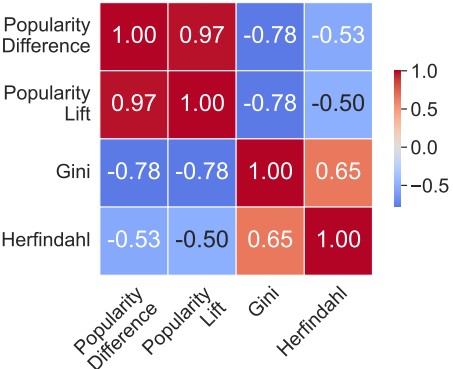

**Figure 4: Kendall's Tau correlation coefficients between the various metrics measured across experiments reported in Tables 2 and 3.**

similar automatic prompt optimization techniques could also be applied to find more balanced prompt-based mitigation strategies that find a happy medium between the two strategies we experimented with.

An open research question is the evaluation of intent-based performance of LLM-based recommenders. As discussed in Section 2.2,

easily generating recommendations that are guided by user-intent is a compelling promise of LLM-based RS. In order to evaluate the performance in this task, one would require structured ground truth, or, as a starting point, use simple proxy tasks where the intent encodes known metadata such as director, genre, year or similar. Similarly, while research in popularity bias is motivated by "rich-get-richer" effects over time, we have not modeled or measured temporal dynamics (see [32, 38, 57] for dynamical modeling approaches, and [49] for an overview).

## 6 CONCLUSION

We have investigated the promise of LLM-based recommender systems focusing on a specific aspect of their performance and usability: popularity bias. We started by constructing a measurement for the phenomenon. To do so, we have formulated desiderata for a popularity bias metric, and have evaluated existing metrics against these desiderata. Applying some adjustments to the Popularity Lift metric by Abdollahpouri et al. [2], we have arrived at our metric: log popularity difference. We acknowledge that this metric may not suit all future studies on popularity bias given the topic's diverse application domains and goals. However, we encourage future research to examine the assumptions and theoretical properties of their chosen metrics, with our framework potentially serving as a useful starting point.

Using our metric, we have compared traditional RS against the LLM-based models on the MovieLens 10M dataset. We have found the LLM-based recommenders to have moderate amounts of popularity bias; usually less than their traditional, collaborative filtering-based counterparts. In our mitigation experiment, we have found that it is possible to lower popularity bias further by including additional instructions in the prompt. In the extreme case, where the model is specifically instructed to 'avoid mainstream blockbusters', we achieve negative popularity bias. This is accompanied by a drop in recommendation accuracy, which overall is lower for our naïve LLM-based recommenders compared to collaborative filtering baselines.

We believe that LLM-based recommender systems will see widespread usage. We encourage practitioners to measure the popularity bias of such models, and, especially in light of its simplicity, experiment with popularity-debiasing via prompting before deploying such a system.

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

# A IMPLEMENTATION DETAILS

## A.1 Implementation details: LLM-based WOK recommender

We used OpenAI's `gpt-3.5-turbo-0613` ("gpt-3.5" in the main text) and `gpt-4-1106-preview` ("gpt-4" in the main text) APIs, as well as Anthropic's `claude-instant-1.2` ("claude-v1" in the main text) and `claude-2.1` ("claude-v2.1" in the main text) APIs. All LLMs used a temperature parameter of 0.0, a *top-p* parameter of 1 (default for both OpenAI and Anthropic APIs), and a *top-k* parameter of 250 (default). The base prompt template used for all experiments is given in Code Listing 1.

```
"""
You are a helpful movie-expert AI tasked with recommending a collection of movies based on a
    user's watch history. The user has watched the following movies in the past:
{watch_history}

# Output instructions
- Immediately start with the movies. Do not provide an introduction.
- Provide a list of {nr_items} movies.
- For each movie, start a new line, indicate the position in the movie list (that is, 1., 2.,
    ...).
- Name the title of the movie (without quotation marks!) and then in parentheses the release
    year.
- Do not recommend movies that the user has already watched. Those are the ones listed above.
- Do not recommend movies that are newer than 2008.

Now create the movie list!"""
```

**Code Listing 1: Prompt template used for the LLM movie recommender. The placeholder `watch_history` is replaced by a list of movies watched by the user at runtime.**

# B ON THE DIFFERENCE OF AVERAGE POPULARITY LIFT AND LOG POPULARITY DIFFERENCE AND THE IMPORTANCE OF STATISTICAL WELL-BEHAVEDNESS (DESIDERATUM 1).

We have seen that average popularity lift and log popularity difference are strongly correlated (Section 4.6), but differ in that average popularity lift does not satisfy the Desiderata (1), (3) and (4), see Table 1. We have motivated the importance of these desiderata theoretically in Section 3.1, and investigate their practical implications here, specifically for well-behavedness (1). Figure 5 illustrates the cumulative averages of popularities $\sum_{i+1}^{N} \phi(a_i)/N$ for $N$ a varying number of observations ($x$-axis) under average popularity lift (left) and log popularity difference (right). For the average popularity lift metric, there is a sudden jump when the largest popularity item is added (around $i = 7500$). This results from the Power-law distribution of unnormalized scores ($g$ =identity, Table 1) and the resulting ill-behavedness. As a consequence, average popularity lift measurements will be highly sensitive to whether or not individual, high popularity items are included in recommendation or user sets $p_u$ or $p_r$, respectively, and will measure any differences in the low-mid popularity regime less reliably.

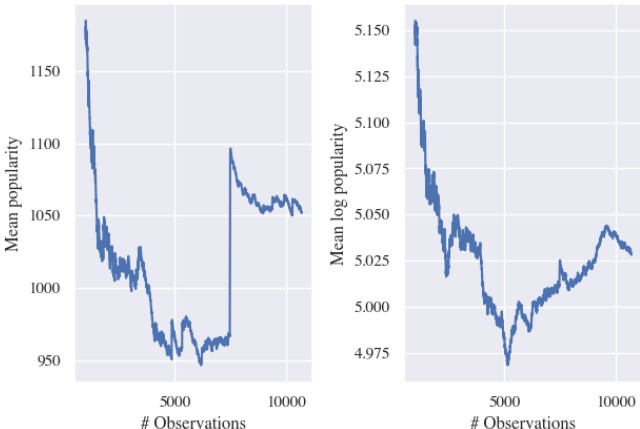

Figure 5: Behavior of the average popularity bias as a function of the data points.