# OpenReview forum: "Large Language Models as Recommender Systems: A Study of Popularity Bias"
_ACM.org/SIGIR/2024/Workshop/Gen-IR — Gen-IR_SIGIR24_

### Official Review · Reviewer_7PZf · 2024-05-27
**A nice read with interesting empirical insights and relevant topics for discussion**

**Rating:** 2
**Confidence:** 4

**Review:**

The paper addresses two very relevant open areas in recommender systems, as are popularity and leveraging LLMs. The paper provides interesting ideas and insights, that are definitely worthy of being brought to discussion in the workshop.

Measuring the popularity bias of LLM-generated recommendations is one interesting data point I find in the paper. It seems to be generally assumed that LLMs are expected to bias towards popular recommendations, but it is interesting to contrast that with data, as this paper does - to find a quite nuanced version of the common belief.

In the observations, the LLM-generated recommendations' HR is an order of magnitude lower than simple kNN baselines. It may be unclear how useful those recommendations can be, and at this level, it shouldn't be difficult to achieve low popularity; the challenge lies in achieving a good tradeoff. Still, to be fair, the LLMs in turn perform an order of magnitude above top pop, which is itself one meaningful ranking any recommendation site or app includes.

I also read the discussion of popularity bias metrics with interest, even though I did not find a large extent of proper novelty. E.g. average log popularity is not a new metric and has been used in a fair amount of highly cited literature - perhaps reversed and referred to as "novelty". The high correlation between log and not-logged popularity, and low correlation to gini, as well as connections to other novelty and diversity metrics have been likewise studied before. Nevertheless, I think this problem space is worthy of further attention as this paper grants.

A couple of lesser details:
- Power-law is usually an idealized assumption for empirical long-tailed distributions. Interaction data rarely conforms literally to a power-law.
- Why do you not use MovieLens 1M instead of subsampling 10M?

---

### Official Review · Reviewer_Vcpk · 2024-05-28
**Interesting work but has limitation in experiments**

**Rating:** 1
**Confidence:** 4

**Review:**

In this paper, the authors study the popularity bias problem in LLM-based recommender systems. The problem itself is very interesting and important, given that LLMs are trending tools and their applications in recommender systems have been extensively studied. The fairness and bias problem in recommender systems has also been well studied. Based on the above aspects, studying the popularity bias and investigating its difference from that in traditional recommender systems is valuable.

The authors reveal that LLM-based recommender includes less popularity bias, evaluated by a new metric proposed by the authors. The established assumption on how popularity bias is measured is interesting. The paper is generally well-written and easy to read.

One of the main contributions of the paper is investigating the difference between traditional and LLM-based recommender systems. The main problem of the paper is that only some simple versions of two types of systems are compared, so it is hard to confirm whether the conclusion is reliable. It would be great if more experiments could be conducted, with more baselines and datasets.

---

### Decision · Program_Chairs · 2024-05-30

**Decision:**

Accept

**Comment:**

This work investigates the impact of popularity bias on recommendations from LLMs using a proposed approach for measuring this. Reviewers found this to be an interesting and important topic, though they pointed out some limitations with the experiments and with how the proposed metric compares to prior work.